# Program Synthesis via Test-Time Transduction

**Kang-il Lee**[1]    **Jahyun Koo**[2]    **Seunghyun Yoon**[3]    **Minbeom Kim**[2]
**Hyukhun Koh**[2]    **Dongryeol Lee**[1]    **Kyomin Jung**[1,2*]
[1]Dept. of ECE, Seoul National University    [2]IPAI, Seoul National University
[3]Adobe Research
{4bkang,kjung}@snu.ac.kr

## Abstract

We introduce transductive program synthesis, a new formulation of the program synthesis task that explicitly leverages test inputs during synthesis. While prior approaches to program synthesis—whether based on natural language descriptions or input-output examples—typically aim to generalize from training examples, they often struggle with robustness, especially in real-world settings where training examples are limited and test inputs involve various edge cases. To address this, we propose a novel framework that improves robustness by treating synthesis as an active learning over a finite hypothesis class defined by programs' outputs. We use an LLM to predict outputs for selected test inputs and eliminate inconsistent hypotheses, where the inputs are chosen via a greedy maximin algorithm to minimize the number of LLM queries required. We evaluate our approach on four benchmarks: Playgol, MBPP+, 1D-ARC, and programmatic world modeling on MiniGrid. We demonstrate that our method significantly improves program synthesis in both accuracy and efficiency. We release our code at https://github.com/klee972/SYNTRA.

## 1 Introduction

Program synthesis is the task of generating programs from a given specification, where the format of the specification can vary widely depending on the problem setting. Recent approaches to program synthesis using large language models [47, 30] rely on a natural language description, usually accompanied by a few test cases, to produce a program. In inductive program synthesis, the model operates without a natural language description, using only a set of input-output examples [48, 38, 19]. A common strategy in both lines of work involves sampling or enumerating multiple candidate programs and selecting those that satisfy the specification by executing them on the provided training examples. However, relatively little attention has been paid to settings where test inputs are available at synthesis time, i.e., the transductive learning scenario.

Vapnik famously advocated for transductive inference [45] with the principle: *"When solving a problem of interest, do not solve a more general problem as an intermediate step."* In the context of program synthesis, this suggests that full generalization through induction may not be necessary if the goal is to predict outputs for a fixed set of test inputs. Such transductive scenarios are common in real-world applications such as spreadsheet automation or data transformation, where the goal is to synthesize a *one-off* program that correctly completes a given set of test inputs (Figure 1). In these settings, the number of training examples is often limited, as they are typically filled manually by users. As a result, programs synthesized from few examples may lack robustness when applied to the test inputs, especially if those inputs include edge cases (i.e., inputs that are atypical compared to the training examples or expose corner-case bugs in program logic). This limitation arises from

---

*Corresponding authors.

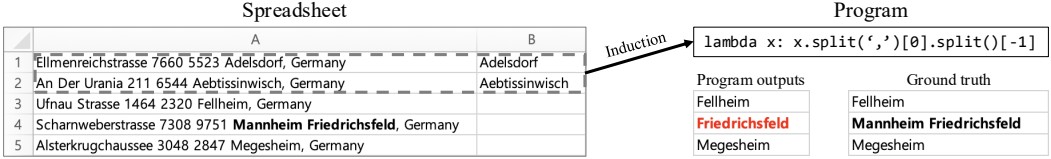

Figure 1: An example of transductive program synthesis. Given the training examples (rows 1 and 2) as input, the inductive program synthesizer generates a program that satisfies these examples. However, this program produces an **incorrect output** for the test input in row 4, which represents an **edge case**.

*epistemic uncertainty*; the model is uncertain about what kinds of inputs will appear at test time. To address this, we introduce **transductive program synthesis**: an approach that explicitly leverages the available test inputs during synthesis to reduce uncertainty and produce more robust programs.

In this work, we formalize transductive program synthesis and propose **SYNTRA** (**SYN**thesis-by-**TRA**nsduction) framework to improve the robustness of programs. The input to transductive program synthesis consists of a program specification and a set of $N$ test inputs, and the goal is to synthesize a program that produces correct outputs on those test inputs. A straightforward approach to this problem might be to feed an LLM with the specification and test inputs and then either ask the model to (1) generate a program that satisfies them, or (2) directly predict the test outputs. However, both approaches suffer from poor scalability; their efficiency and performance degrade rapidly as the number of test inputs increases.

Alternatively, we approach this problem as a learning over a finite hypothesis class. The hypothesis class $\mathcal{H}$ is defined as a set of $N$-tuples, consisting of program outputs on the test inputs, where the programs are generated by a **program synthesis model** based on the given specification. In our work, we implement the model using an LLM for its capability to generate code in general-purpose languages (e.g., Python). The programs' outputs on test inputs are collected and deduplicated to construct $\mathcal{H}$. We assume a realizable setting, in which there exists at least one hypothesis $h^* \in \mathcal{H}$ that matches all ground truth outputs. The objective is to identify this correct hypothesis $h^*$.

To achieve this, we leverage a **transduction model** that observes a test input and program outputs as candidates, and selects one output as a pseudo-label. Hypotheses inconsistent with this pseudo-label are eliminated from the current version of the hypothesis class. This process of transductive prediction and hypothesis elimination is repeated iteratively until a single hypothesis remains. Here, the number of queries to the transduction model depends on which inputs are queried and in what order. To minimize this cost, we propose a greedy maximin algorithm, which selects the test input that eliminates the largest number of hypotheses in the worst case. We instantiate the transduction model using an LLM, leveraging its reasoning abilities and world knowledge to produce high-accuracy pseudo-labels. As a result, our framework offers the best of both worlds: program synthesis (precision, efficiency and interpretability) and LLMs (common sense and world knowledge).

We evaluate our method on four program synthesis datasets: Playgol [9], an inductive programming benchmark for string transformation, MBPP+ [31], a benchmark for generating code from a natural language description, 1D-ARC [49], a visual reasoning benchmark, and programmatic world modeling on MiniGrid [7] environment. On these benchmarks, our algorithm significantly outperforms purely inductive [26] or transductive [33] methods. Moreover, by choosing test inputs according to the maximin criterion, we achieve comparable accuracy with substantially fewer LLM calls (halving the extra LLM calls above the lower bound) than when selecting inputs at random. We also empirically show that the number of required query increases sublinearly with the number of inputs, making it scalable to large test sets.

Our contributions are as follows:

- We formulate **transductive program synthesis** as a new task.
- We propose **SYNTRA**, a general framework that significantly improves the robustness of program synthesis on edge cases, by leveraging test inputs through a transduction model.
- We instantiate this framework using large language models and evaluate it on four datasets, showing up to 196% improvements in task accuracy.

## 2 Related Work

### 2.1 Program Synthesis with LLMs

Large language models have recently emerged as powerful tools for program synthesis, significantly advancing the automation of software development tasks. Models such as Codex [3] and Code Llama [39] have demonstrated strong performance on benchmarks like HumanEval [3] and MBPP [1].

Several works have explored enhancing program synthesis by using execution feedback to iteratively refine candidate programs [38, 23, 43], and by generating diverse solutions and selecting the best candidate based on test case results [29, 30, 47, 32] or functional consensus [25, 2, 42]. Despite these advances, the reliability of generated programs remains a challenge, particularly in the presence of edge cases or under-specified tasks [31, 4]. Our work seeks to improve robustness in such settings by leveraging available test inputs and the LLM's transductive prediction capability.

### 2.2 Inductive Program Synthesis

Our work on transductive program synthesis is closely related to the extensively studied area of inductive program synthesis. It aims to generate a program from input-output examples, with the synthesized program expected to generalize to unseen inputs. Applications of inductive synthesis include string transformation [17, 11, 22], spreadsheet automation [6], list processing [40], visual reasoning [8, 28], symbolic regression [16], and graphics generation [13].

Early approaches to inductive program synthesis mostly relied on hand-crafted domain-specific languages (DSLs) to limit the space of possible programs [5, 35]. Recently, LLMs have emerged as powerful tools for inductive synthesis tasks, due to their ability to leverage extensive pre-trained knowledge and code generation capabilities in general-purpose languages such as Python [48, 27, 46].

Most of the mentioned works assume scenarios in which the number of training examples is sufficient to uniquely determine a single program. Some studies have explored designing optimal inputs for induction [37, 15] and using direct transductive prediction when program induction fails [28]. While researchers adopt Bayesian program learning [21, 12, 36] to address uncertainty and learning from few examples, its primary focus is learning a prior from training data rather than leveraging multiple test inputs during inference. Our work explicitly makes use of test inputs and proposes an effective methodology for addressing them.

## 3 Transductive Program Synthesis and SYNTRA Framework

We begin by formally defining the task of transductive program synthesis. We then describe the most general form of the Synthesis-by-Transduction (SYNTRA) framework, followed by a detailed explanation in Section 4 of how we instantiate this framework using large language models.

### 3.1 Problem Definition

Our problem formulation closely resembles that of transductive inference. Given an input set $\mathcal{X}$ and an output set $\mathcal{Y}$, consider a function $f^* : \mathcal{X} \to \mathcal{Y}$ with a specification $S$. $S$ includes $M$ train input-output pairs $\{(x_i, y_i)\}_{i=1}^{M} \in (\mathcal{X} \times \mathcal{Y})^M$ where $f^*(x_i) = y_i$ for all $i \in [M]$, and (optionally) a natural language task description $t$. Also, there is a set of $N$ test inputs $\{\tilde{x}_i\}_{i=1}^{N}$ visible to the system. The goal of the task is to predict the test outputs $\{\tilde{y}_i\}_{i=1}^{N} = \{f^*(\tilde{x}_i)\}_{i=1}^{N}$, given $S$ and $\{\tilde{x}_i\}_{i=1}^{N}$.

In transductive program synthesis, predictions for the outputs are made by first synthesizing a program $f$, and then applying it to the test inputs. We expect $f$ to produce correct outputs for the given test inputs; the primary concern here is not the overall correctness or generality of $f$, but rather its accuracy on the specific test set. Nevertheless, producing a predictive model in the form of an executable program offers several advantages, as will be discussed further in Section 6.

### 3.2 Synthesis-by-Transduction (SYNTRA)

We frame the above problem as an active learning problem over a finite hypothesis class.

**Algorithm 1:** SYNTRA

> **Input:** Specification $S$ with training examples $\{(x_j, y_j)\}_{j=1}^{M}$; Test inputs $\{\tilde{x}_i\}_{i=1}^{N}$; Program synthesis model $\sigma$; Transduction model $\tau$
> **Output:** Hypothesis $h^*$

**1 Function** Y$(i, \mathcal{V})$:
**2**    **return** $\{h[i] | h \in \mathcal{V}\}$
**3** $\mathcal{P} \leftarrow \sigma(S)$             // Generate programs
**4** $\mathcal{P}' \leftarrow \{f \in \mathcal{P} | f(x_j) = y_j, \forall j \in [M]\}$     // Filter by training examples
**5** $\mathcal{H} \leftarrow$ exe_dedup$(\mathcal{P}', \{\tilde{x}_i\}_{i=1}^{N})$     // Get execution results and deduplicate
**6** $\mathcal{V}_0 \leftarrow \mathcal{H}$             // Initial version space
**7** $t \leftarrow 0$
**8 while** $|\mathcal{V}_t| > 1$ **do**
**9**    $\mathcal{I} \leftarrow \arg\max_{i \in [N]} \min_{y \in \text{Y}(i, \mathcal{V}_t)} |\{h \in \mathcal{V}_t | h[i] \neq y\}|$    // A set of maxes of mins
**10**    $i^* \leftarrow \arg\min_{i \in \mathcal{I}} \sum_{y \in \text{Y}(i, \mathcal{V}_t)} \text{len}(y)$     // Tie-break by shorter outputs
**11**    $\hat{y} \leftarrow \tau(S, \tilde{x}_{i^*}, \text{Y}(i, \mathcal{V}_t))$        // Transductive prediction
**12**    $\mathcal{V}_{t+1} \leftarrow \{h \in \mathcal{V}_t | h[i^*] = \hat{y}\}$      // Eliminate inconsistent hypotheses
**13**    $t \leftarrow t + 1$
**14 end**
**15 return** $h^* \in \mathcal{V}_t$

**Hypothesis class** The construction of the hypothesis class $\mathcal{H}$ (Alg. 1 L3~L6) follows these steps:

1. Generate a set of $K$ candidate programs $\mathcal{P}$ using a program synthesis model $\sigma$.

2. Filter the programs to retain only those that satisfy all $M$ provided training input-output pairs. This step yields $\mathcal{P}' = \left\{ f \in \mathcal{P} \ \middle| \ \bigwedge_{i=1}^{M} f(x_i) = y_i \right\}$.

3. Execute the programs in $\mathcal{P}'$ on the $N$ test inputs and deduplicate the execution results to construct our hypothesis class $\mathcal{H} = \{(f(\tilde{x}_1), f(\tilde{x}_2), ..., f(\tilde{x}_N)) | f \in \mathcal{P}'\}$. Note that the elements of $\mathcal{H}$ are not programs themselves, but the outputs of those programs.

Since the hypothesis class defined above is only verified against the training input-output pairs, we must select a hypothesis that robustly generalizes the diverse cases that may appear in the test inputs. To this end, we iteratively repeat the process of input query selection, transductive prediction, and hypothesis elimination until only a single hypothesis remains.

**Input query selection** To leverage the power of the transduction model, we must decide which input to query for a prediction. The number of queries required to eliminate all but one hypothesis depends on which inputs are selected and in what order, making this choice a critical component of the method. We select an input based on a criterion that greedily maximizes the number of hypotheses eliminated in the worst case (Alg. 1 L9~L10).

| | $h_1$ | $h_2$ | $h_3$ | $h_4$ | min |
|---|---|---|---|---|---|
| $\tilde{x}_1$ | A | A | B | C | **2** |
| $\tilde{x}_2$ | D | D | D | E | 1 |
| $\tilde{x}_3$ | F | G | G | G | 1 |

Figure 2: An example of the maximin algorithm. The numbers of eliminated hypotheses in the worst case are shown in the "min" column.

To describe what our maximin criterion does: for each input, we first consider the worst-case prediction by the transduction model—that is, the scenario in which the prediction eliminates the fewest hypotheses (as illustrated in the "min" column of Figure 2). We then select the input for which this minimum number of eliminated hypotheses is maximized ($\tilde{x}_1$ in Figure 2).

Let us denote the $i$-th element of $h$ as $h[i]$, and the deduplicated output set for input $\tilde{x}_i$ and hypothesis class $\mathcal{H}$ as $\mathcal{Y}_{i, \mathcal{H}}$. In other words, $\mathcal{Y}_{i, \mathcal{H}} = \{h[i] | h \in \mathcal{H}\}$. Then our proposed criterion to select the input index $i^*$ can be represented as follows.

$$i^* = \arg\max_{i \in [N]} \min_{y \in \mathcal{Y}_{i, \mathcal{H}}} |\{h \in \mathcal{H} | h[i] \neq y\}| \tag{1}$$

While this approach does not guarantee a globally optimal solution, it can be seen as a greedy algorithm that makes a locally optimal decision at each iteration. If we can assume that each query eliminates at least a certain fixed proportion of hypotheses, then this approach requires $O(\log |\mathcal{H}|)$ queries. Similar query selection mechanisms are widely used in the active learning literature [10, 34], where it is well understood that outperforming such greedy algorithms is often provably hard [18] in the absence of additional information.

When a tie occurs in the maximin value, we break ties by selecting the input whose set of possible output candidates has the shortest total length. This reduces the length of the input passed to the transduction model in the next step, helping to reduce computational cost and alleviate reasoning burden.

**Transductive prediction**    The next step is to use the transduction model $\tau$ to predict the output for the selected input (Alg. 1 L11). Presumably, the transduction model is implemented using an LLM, due to its extensive world knowledge acquired from vast corpora and strong reasoning capabilities. The model's input consists of the specification $S$, selected test input $\tilde{x}_{i^*}$ for which the output is to be predicted, and the set of candidate outputs $\mathcal{Y}_{i^*, \mathcal{H}}$. The model's output $\hat{y}$ is one of the elements from the candidate output set.

$$\hat{y} = \tau(S, \tilde{x}_{i^*}, \mathcal{Y}_{i^*, \mathcal{H}}) \tag{2}$$

**Hypothesis elimination**    As the final step of each iteration, we eliminate all hypotheses that are inconsistent with the output predicted by the transduction model (Alg. 1 L12). We define the **version space** at iteration $t$, denoted as $\mathcal{V}_t$, as the set of hypotheses consistent with all training and test observations collected up to iteration $t$.

$$\mathcal{V}_t = \{h \in \mathcal{V}_{t-1} | h[i^*] = \hat{y}\}, \mathcal{V}_0 = \mathcal{H} \tag{3}$$

## 4    LLM-Based Instantiation

So far, we have described the most general form of the SYNTRA framework. In this section, we provide details on how we instantiate this framework using an LLM, focusing on the implementation of the program synthesis model $\sigma$ and the transduction model $\tau$.

### 4.1    Program Synthesis Model

In our work, the program synthesis model $\sigma$ is a function that takes program information as input and generates a set of candidate programs. We implement this model by prompting an LLM. The simplest approach is providing the LLM with a natural language instruction and program specification, and then obtaining multiple candidate programs through repeated IID sampling. A crucial consideration here is the *semantic diversity of the generated programs*, as diversity directly influences the expressiveness of the hypothesis space and thus significantly impacts the final system performance. However, IID sampling from the most powerful LLMs available today often results in programs with limited semantic diversity [24].

To overcome this limitation, we prompt the LLM to first generate distinct algorithms (implementations) for solving the given programming task. Subsequently, we prompt the LLM to translate each algorithm into executable Python code. By generating algorithm lists of length $c$ through $s$ rounds of IID sampling, we ultimately obtain a total of $cs$ candidate programs as $\mathcal{P}$. In Appendix C.2, we observe that this approach indeed boosts diversity, leading to an increased number of tasks for which at least one correct program is generated. Henceforth, we refer to this approach as **AGA** (Autoregressively Generated Algorithms).

### 4.2    Transduction Model

In our framework, the role of the transduction model $\tau$ is to predict the output corresponding to a given input. The choice of LLM for implementing the transduction model is presumed to be more capable than the one used for the program synthesis model. This is because the program synthesis model needs to generate multiple candidate programs, making computation the main bottleneck. In contrast, the transduction model is expected to be called fewer times, but each prediction must be highly accurate. We instantiate this model using two types of LLMs (gpt-4.1-2025-04-14 or

`gpt-4o-mini-2024-07-18`) and compare their performance in Section 5.2. Specifically, the LLM is instructed to predict the correct output for a given test input, conditioned on the specification and candidate outputs. Additionally, we use zero-shot chain-of-thought prompting [20] to encourage explicit reasoning by the LLM. Since the LLM's output is not guaranteed to exactly match one of the candidate outputs, we use fuzzy string matching to select the candidate that is most similar to the LLM's prediction. We set the temperature of the program synthesis model to 1 and that of the transduction model to 0.7. Detailed prompts for both models are in Appendix B.

# 5 Experiment

## 5.1 Dataset

| Playgol | MBPP+ | 1D-ARC | MiniGrid |
|---|---|---|---|
| "Conger, Minnesota(MN)"
↓
"State: Minnesota" | Write a function to check whether a list contains the given sublist or not. | [0,0,1,0,0,0,1,0]
↓
[0,0,1,1,1,1,1,0] | Wall(0,0); Goal(1,0);
Wall(0,1); Agent(1,1,direction=(1, 0))

↓ action: turn left |
| "Princeton, New Jersey(NJ)"
↓
"State: New Jersey" | [[3, 5, 7], [3, 7]] → False
[[4, 3], [4, 3]] → True
[['r'], []] → True | [0,1,0,0,1,0,0,0,0]
↓
[0,1,1,1,1,0,0,0,0] | Wall(0,0); Goal(1,0);
Wall(0,1); Agent(1,1,direction=(0, -1)) |

Figure 3: Examples of Playgol, MBPP+, 1D-ARC and MiniGrid domain. Test outputs are highlighted in green.

We apply transductive program synthesis to four domains: string transformation, Python programming, visual reasoning, and programmatic world modeling (Figure 3). The string transformation domain is central to spreadsheet automation technologies, such as FlashFill [17] and Smart Fill [14]. Among the available datasets, we select Playgol[2] [9], a real-world dataset originally designed for inductive programming, as our benchmark.[3] In Playgol, the original task is to generate a program consistent with a set of given input-output examples. Each task in Playgol provides five input-output examples; to simulate realistic conditions involving epistemic uncertainty, we use only one example as a training example and treat the remaining four examples as test inputs.

For the Python programming domain, we use the MBPP+ dataset [31] to evaluate our methodology. Compared to MBPP [1], MBPP+ provides significantly more diverse and numerous test cases, making it especially suitable for evaluating our framework, which assumes many available test inputs and potential edge cases. Furthermore, MBPP+ provides natural language instructions describing the desired functionality for each task. This setting mirrors realistic scenarios where a user provides input data along with an instruction specifying the task to be performed. MBPP+ provides at least 52 input-output pairs for every task; we utilize one example as training data and between 5 and 50 examples as test cases.

In the visual reasoning domain, we use 1D-ARC [49]. 1D-ARC is a 1D version of the challenging 2D grid visual reasoning benchmark, ARC [8], and it includes a variety of visual concepts (e.g., fill, flip, mirror, denoise, etc.). In this benchmark, we use 1 example as the training set and 3 examples as the test set.

Finally, we validate SYNTRA's ability on programmatic world modeling (e.g. WorldCoder [44])–a complex task that requires modeling interaction mechanisms between different entities and actions. We used two MiniGrid [7] environments (DoorKey, UnlockPickup), and focused on generating a transition function that, given the current state and action, outputs the next state. In our experiment, the synthesis model receives the current state, action list, and natural language mission as input, and generates the world models. The transduction model's role is to select the most plausible next state candidate among multiple world model predictions. For evaluation, given a state and an action, the world model selected by SYNTRA predicts the next state, which we then compare to the ground truth next state. The state and action pairs for evaluation are collected from human play. This task is well-suited for transductive program synthesis, as the action space is typically known beforehand and can serve as a visible test input. Since programmatic world modeling differs in nature from the three domains discussed earlier, we present it separately in Section 5.4.

---

[2]The name "Playgol" originally refers to an inductive logic programming system [9]. We use the name here to refer to the string transformation dataset introduced in that work.

[3]We manually corrected some mislabeled tasks of the dataset.

## 5.2 Main Results

Our primary focus in this section is the learning over the hypothesis class defined earlier. Therefore, we filter out tasks where learning is trivial. Specifically, we only retain tasks where the hypothesis class $\mathcal{H}$ constructed by $\sigma$ contains both correct and incorrect candidate hypotheses. After this filtering, we obtain 119 tasks with 4 test inputs from Playgol, 149 tasks with 50 test inputs from MBPP+, and 124 tasks with 3 test inputs from 1D-ARC for our evaluation.

Table 1: Comparison of different approaches on the filtered Playgol and MBPP+ datasets. Filtering is based on the 32 programs generated using AGA ($c = 4, s = 8$) with `gpt-4o-mini-2024-07-18`.

| Approach | **Playgol** (1 train / 4 test) | | | **MBPP+** (1 train / 10 test) | | |
|---|---|---|---|---|---|---|
| | Task Acc. | Example Acc. | # $\tau$ Calls | Task Acc. | Example Acc. | # $\tau$ Calls |
| Random program $f \in \mathcal{P}'$ | 66.6 | 79.9 | - | 70.6 | 88.2 | - |
| Random hypothesis $h \in \mathcal{V}_0$ | 37.6 | 62.7 | - | 43.4 | 76.8 | - |
| *gpt-4.1 for $\tau$* | | | | | | |
| LLM direct transduction [33] | 85.7 | 93.7 | 476 | 59.7 | 87.2 | 1490 |
| SYNTRA w/ random query | **93.3** | 96.0 | 144 | 84.6 | 94.1 | 198 |
| SYNTRA w/ maximin | **93.3** | **96.3** | **131** | **85.9** | **95.6** | **164** |
| *gpt-4o-mini for $\tau$* | | | | | | |
| LLM direct transduction [33] | 72.3 | 87.4 | 476 | 35.6 | 75.0 | 1490 |
| SYNTRA w/ random query | 91.6 | 95.5 | 140 | **75.2** | **90.4** | 190 |
| SYNTRA w/ maximin | **93.3** | **96.3** | 132 | 73.2 | 89.5 | **163** |

In Table 1 and 2, we evaluate our proposed methodology against several baselines. In this experiment, we use 10 test inputs out of 50 for MBPP+. We report two primary accuracy metrics: task-level accuracy, defined as the percentage of tasks for which all test outputs are predicted correctly, and example-level accuracy, defined as the proportion of correctly predicted test outputs. Additionally,

Table 2: Comparison of different approaches on the filtered 1D-ARC dataset. Filtering is based on the 128 programs generated using MoC [26] with `gpt-4.1-mini-2025-04-14`.

| Approach | **1D-ARC** (1 train / 3 test) | | |
|---|---|---|---|
| | Task Acc. | Example Acc. | # $\tau$ Calls |
| Random program $f \in \mathcal{P}'$ | 24.0 | 28.7 | - |
| *gpt-4.1 for $\tau$* | | | |
| LLM direct transduction [33] | 41.9 | 68.1 | 372 |
| SYNTRA w/ random query | **71.8** | **82.1** | 179 |
| SYNTRA w/ maximin | **71.8** | 80.8 | **159** |

we report the number of transduction model calls as a measure of efficiency (see Section 6 for more detailed discussion on computational cost). We compare SYNTRA with the following ablations.

- **Random program** selects a program uniformly at random from the set of candidates that are consistent with the training example (i.e., from $\mathcal{P}'$ in our algorithm).

- **Random hypothesis** first deduplicates the outputs of the programs to form a hypothesis class, then samples a single hypothesis uniformly at random from this set. This baseline performs significantly worse than the random program baseline, suggesting that correct programs are sampled more frequently before output-based deduplication.

- **LLM direct transduction** bypasses program synthesis entirely and instead asks the LLM to directly predict test outputs given the training example and test inputs. The prompt explicitly instructs the LLM to reason step-by-step. Interestingly, this approach outperforms the synthesis baseline (random program) on Playgol and 1D-ARC but underperforms on MBPP+. We attribute this to the fact that Playgol and 1D-ARC tasks often benefit from world knowledge and pattern recognition (a strength of LLMs), whereas MBPP+ tasks tend to be more algorithmic in nature (a strength of programs). A key limitation of direct LLM transduction is that the number of LLM calls scales linearly with the number of test inputs, making the method computationally impractical when the test set is large.

- **SYNTRA with random query** is a variant of SYNTRA, which randomly selects the input query (from those with at least two possible output candidates) as an ablation of the maximin criterion. As shown in the table, this approach already yields significant improvements over all baselines in both domains and with both models. The performance gain is especially pronounced when using a more capable model like `gpt-4.1`.

- **SYNTRA with maximin criterion** is the full version of our method, including the maximin input selection criterion. Compared to the random query variant, this method substantially reduces the number of transduction model calls, particularly in MBPP+, where the number of test inputs is larger. This result highlights the efficiency and scalability of our SYNTRA framework.

Appendix C.1 presents additional experimental results using smaller open-source LLMs. Additionally, Appendix D provides examples where our methodology succeeds and fails, along with an analysis of its strengths and weaknesses.

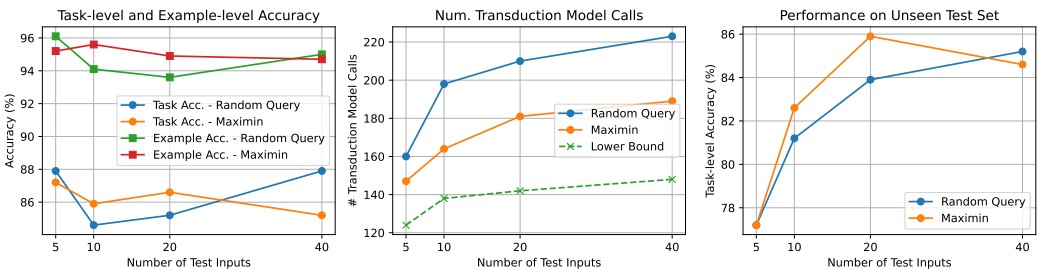

Figure 4: Experimental results on test input scaling and the unseen test set.

**Scaling test set size**    In real-world applications, program synthesis is especially valuable when the number of test inputs is so large that manual processing is cumbersome. To simulate such conditions, we leverage the abundant test cases in MBPP+. Specifically, we vary the number of visible test inputs provided to the system: 5, 10, 20, and 40. For each setting, we measure accuracy and the number of transduction model calls.

In Figure 4, we observe that task-level and example-level accuracy remain relatively stable regardless of the number of test inputs. In terms of transductive model call counts, the number of calls increases **sub-linearly** with the number of test inputs, demonstrating the scalability of our framework. Notably, the gap between the maximin and random query methods also widens as the test set grows. In addition, we indicate a "lower bound," the number of tasks where the initial hypothesis class $\mathcal{V}_0$ contains at least two distinct hypotheses. This value serves as a rather conservative lower bound on the number of necessary LLM calls, since at least one query is needed to resolve any ambiguity (the true minimum number of calls is likely higher). When comparing the number of LLM calls to this lower bound, we find that the maximin criterion approaches roughly twice the efficiency of the random query strategy. These results indicate that the maximin algorithm significantly improves the efficiency of the SYNTRA framework.

**Performance on unseen test set**    Next, we investigate how well the programs synthesized via SYNTRA generalize to an unseen test set. The unseen test set is constructed by selecting 10 test inputs from the 50 available in MBPP+, while the remaining 40 are used as test inputs visible to the system. We run SYNTRA using 5, 10, 20, or 40 of these visible test inputs and measure the task-level accuracy of the resulting program on the unseen test set.

The results in Figure 4 show a general trend: as the number of visible test inputs increases, the accuracy on the unseen test set also improves. When using 20 or more visible test inputs, the accuracy on the unseen test set approaches the task accuracy reported in Table 1. This suggests that programs synthesized via SYNTRA from a sufficiently large number of test inputs can be expected to perform comparably well even on new, unseen inputs.

### 5.3   Variations on Program Synthesis Model

Up to this point, we have focused on how efficiently and robustly our method can select a correct program from fixed hypothesis class constructed by particular synthesis model. However, when considering expected performance over all tasks, the choice of synthesis model becomes critically important. In this section, we examine how the choice of synthesis model affects overall performance on the full, unfiltered datasets. In this experiment, we use 4 test inputs for Playgol, 50 test inputs for MBPP+, and 3 test inputs for 1D-ARC.

**IID sampling**   The AGA approach we use first generates algorithms autoregressively and then translates each into Python code. As a result, the resulting programs do not strictly follow the LLM's output distribution. This deliberate "flattening" of the output probability boosts diversity, which benefits in more challenging tasks [26, 47, 30]. However, it may reduce the likelihood of sampling a correct program in easier tasks, where the correct solution is already highly probable under the model's natural distribution. To investigate this phenomenon in the context of our work, we consider a more standard approach for the synthesis model: IID sampling of programs from a fixed prompt. We evaluate how this affects end-to-end performance.

Table 3 (IID) shows the performance of randomly selecting a program from those obtained via IID sampling. On Playgol and 1D-ARC, IID slightly outperforms AGA. However, in MBPP+, performance drops significantly. This suggests that MBPP+ tasks benefit more from the diversity encouraged by AGA, which increases the chance of synthesizing a robust program.

Table 3: Task accuracies (%) of various approaches.

| Approach | Playgol | MBPP+ | 1D-ARC |
|---|---|---|---|
| AGA | 72.7 | 64.8 | 23.9 |
| AGA + SYNTRA | 82.5 | 72.4 | 37.8 |
| IID | 76.9 | 56.9 | 25.0 |
| IID + SYNTRA | 82.5 | 59.1 | 38.9 |
| MoC [26] | 78.1 | 71.4 | 16.7 |
| MoC + SYNTRA | **83.7** | **74.0** | **49.4** |
| AGA + test inputs as prompt | 80.4 | 63.5 | - |
| AGA + test inputs as prompt + SYNTRA | **84.6** | **70.3** | - |
| IID + test inputs as prompt | 83.2 | 49.3 | - |

Interestingly, the performance gap between AGA and IID on Playgol disappears when we apply SYNTRA (AGA + SYNTRA v.s. IID + SYNTRA). This indicates that AGA did generate the correct program, but it was underrepresented in the overall program pool and thus unlikely to be selected—SYNTRA was able to recover it. In contrast, on MBPP+, applying SYNTRA does not close the gap between AGA and IID, implying that IID sampling failed to generate the correct program at all, leaving no opportunity for SYNTRA to recover it. These observations underscore the value of diversity-enhancing strategies like AGA, especially when combined with effective verification mechanisms like SYNTRA.

**Advanced model**   We also examine the impact on final performance when SYNTRA is applied to state-of-the-art program synthesis model. We use Mixture of Concepts (MoC) [26], a recent inductive program synthesis approach based on LLMs. MoC first generates distinct elementary concepts that may help solve the problem, then produces natural language hypotheses based on these concepts, and synthesizes Python programs based on the hypotheses. For MBPP+, we made a minor modification by including the natural language task description in the prompt.

The results in the table show that MoC alone yields mixed outcomes depending on the benchmark. However, when combined with SYNTRA, performance improves even further, outperforming all other approaches we compared. This demonstrates that the SYNTRA framework can be layered on top of existing strong program synthesis models to push performance beyond current limits.

**Test inputs as prompt**   A straightforward way to directly improve the output distribution of an LLM-based program synthesis model is to include test inputs in the prompt, explicitly instructing the model to generate a program that generalizes to those inputs. While intuitive, this approach is not scalable, as the prompt length increases proportionally with the number of test inputs.

As shown in Table 3 (test inputs as prompt), this method can indeed be beneficial in cases like Playgol, where the number of test inputs is relatively small. However, for MBPP+, including test inputs in the prompt leads to a performance drop for both AGA and IID. This likely results from the excessive prompt length—incorporating all 50 test inputs may overwhelm the LLM and hinder its reasoning ability. These limitations further highlight the importance of scalable alternatives such as SYNTRA, which can robustly select correct programs without overloading the prompt.

## 5.4   Programmatic World Modeling on MiniGrid

Finally, we apply SYNTRA to programmatic world modeling on MiniGrid. Both the synthesis and transduction models are `gpt-4.1-mini-2025-04-14`. We sample 16 IID programs per state, and used example-level accuracy to compute transition function accuracy. In Table 4, SYNTRA shows substantial benefit for the world model synthesis task as well. SYNTRA enables learning a more accurate world model, which would likely result in more efficient planning or policy learning.

Table 4: Task accuracies (%) on MiniGrid.

| Approach | DoorKey | UnlockPickup |
|---|---|---|
| IID | 57.1 | 62.9 |
| IID + SYNTRA | **68.8** | **67.6** |

A good example that illustrates how SYNTRA helps in this task is coordinate notation. In the MiniGrid state representation we used, the positive directions are to the right and downward, and this sign convention can be inferred from the coordinates of surrounding objects. Since this convention is not obvious at the outset, actions such as `turn left` or `turn right` are not always implemented correctly during the synthesis stage. However, the transduction model, by directly observing the candidate output states, was able to identify the correct one.

## 6  Discussion

**Extension to online learning and human-in-the-loop**  Our methodology naturally extends to online or human-in-the-loop settings. After identifying a final hypothesis through SYNTRA, we can retain the corresponding program and, when a new input arrives, detect behavioral divergence across candidate programs. In such cases, the system can invoke the transduction model to update the version space accordingly. Moreover, in situations where the transduction model's confidence is low, the system can selectively ask the user for label, enabling interactive program synthesis with minimal human intervention.

**Transductive program synthesis v.s. LLM direct transduction**  In our experiments, we compared transductive program synthesis and LLM direct transduction primarily by measuring the number of transduction model calls. When considering the full pipeline, the program synthesis method includes a preliminary step of generating 32 candidate programs. In such cases, direct LLM prediction may result in fewer total calls. However, in our experimental setup, we used a smaller model for synthesis and a larger model for transduction, prioritizing prediction quality over generation cost. SYNTRA typically required no more than three calls per task, making the overall cost lower for SYNTRA despite the initial synthesis step. Furthermore, as the number of test inputs increases, the cost of direct transduction increases linearly, whereas our method remains more stable.

Beyond efficiency, transductive program synthesis offers significant advantages over direct transduction in terms of performance, interpretability, and extensibility to online or human-in-the-loop workflows. In domains where some tasks are inherently difficult to express through code, a hybrid approach that ensembles program synthesis with direct prediction may be more effective [28].

**Probabilistic perspective**  Rather than performing probabilistic inference over programs directly, our approach constructs a hypothesis class by deduplicating execution results and eliminates hypotheses based on transductive predictions. This design choice is intended to ensure broad applicability, even to models where program probabilities are difficult to estimate, such as black-box LLMs or synthesis models based on enumerative search. If such probabilities were available, our framework could incorporate probabilistic strategies. For instance, instead of maximin criteria, we could adopt uncertainty-based strategies or more sophisticated methods like query-by-committee [41]. These directions offer promising extensions for future work.

## 7  Conclusion

We introduced transductive program synthesis, a new framework that leverages test inputs during synthesis to improve the robustness and efficiency of program generation. By combining LLM-based program synthesis with transductive prediction and hypothesis elimination, our SYNTRA framework significantly outperforms baselines in terms of accuracy and efficiency. This framework is scalable, interpretable, and extensible; offering a promising direction for robust real-world program synthesis.

## Acknowledgments and Disclosure of Funding

This work was partly supported by Institute of Information & communications Technology Planning & Evaluation (IITP) grant funded by the Korea government(MSIT) (No.RS-2022-II220184, Development and Study of AI Technologies to Inexpensively Conform to Evolving Policy on Ethics, 60%), and Institute of Information & communications Technology Planning & Evaluation (IITP) grant funded by the Korea government(MSIT) [NO.RS-2021-II211343, Artificial Intelligence Graduate School Program (Seoul National University) & NO.RS-2021-II212068, Artificial Intelligence Innovation Hub (Artificial Intelligence Institute, Seoul National University)]. K. Jung is with ASRI, Seoul National University, Korea.

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

# A  Limitation

While SYNTRA demonstrates strong performance, scalability, and explainability across multiple domains, it is important to recognize its limitations.

First, our approach relies on the assumption that visible test inputs exist. This assumption is critical for enabling the transduction model to evaluate and filter candidate programs. In domains where such inputs are absent or unobservable, the method becomes less applicable. However, this limitation can be partially addressed by generating test inputs with the LLM.

Second, SYNTRA is less effective in settings where inputs are semantically meaningless. In such cases, the LLM cannot effectively exploit its prior world knowledge, limiting the benefits of our framework.

Third, although SYNTRA can select the optimal program from a mixture of correct and incorrect candidates, it does not inherently improve the synthesis of highly complex programs. For problems that require deep search or reasoning, the synthesis step remains a bottleneck.

Finally, because LLMs are used as transduction models, undesirable biases present in the models may propagate to the final outputs. This raises concerns about fairness, safety, and robustness.

# B  Prompts

Here, we present the prompts used for our program synthesis and transduction models. The prompts below are all designed for use on Playgol. For MBPP+, we additionally prepended the natural language task description directly before the input-output examples.

---

**Prompt for Program Synthesis Model - Algorithm Generation**

```
You will be given a list of input-output pairs. There are
    multiple algorithms that transform each input to the
    corresponding output.
Generate 4 algorithms for the transformation in natural language
     form.
These algorithms should be distinct; they map the given input to
     the output but implemented in various ways.

Please format your algorithms as follows:

{{
1: "algorithm",
2: "algorithm",
...
}}

Input-output pairs:
{INPUT_OUTPUT_PAIRS}

Algorithms:
```

---

**Prompt for Program Synthesis Model - Code Generation**

```
You will be given a list of input-output pairs and an algorithm
    described in natural language.
Implement the given algorithm in a Python function 'fn' that
    maps the following inputs to their corresponding outputs.

Please format your Python function as follows:

'''python
def fn(x):
    # x is {INPUT_FORMAT}
    # Your code here
    return y # y is {OUTPUT_FORMAT}
'''

Input-output pairs:
{INPUT_OUTPUT_PAIRS}

Algorithm:
{ALGORITHM}

Python function:
```

**Prompt for Transduction Model**

```
Based on given input-output pairs, select which of the outputs
    is most plausible for given test input.
Think step-by-step and enclose your answer with ''' at the end
    of your response.

Input-output pairs:
{INPUT_OUTPUT_PAIRS}

Test input:
{TEST_INPUT}

Test output candidates:
{TEST_OUTPUT_CANDIDATES}
```

# C Additional Results

## C.1 Results with Additional LLMs

In this section, we present experimental results on Playgol using more smaller open-source models. Specifically, we used `Llama-3.1-8B-Instruct` as the program synthesis model and `Llama-3.1-70B-Instruct` as the transduction model.

Table 5: Comparison of different approaches on the filtered Playgol dataset consisting of 124 tasks. Filtering is based on the programs generated using AGA.

| Approach | Playgol (1 train / 4 test) | | |
|---|---|---|---|
| | Task Acc. | Example Acc. | # $\tau$ Calls |
| Random program $f \in \mathcal{P}'$ | 62.5 | 74.8 | - |
| Random hypothesis $h \in \mathcal{V}_0$ | 34.9 | 56.0 | - |
| *Llama-3.1-70B-Instruct for $\tau$* | | | |
| LLM direct transduction [33] | 58.1 | 84.3 | 476 |
| SYNTRA w/ random query | 71.0 | 81.5 | 140 |
| SYNTRA w/ maximin | **78.2** | **84.8** | **128** |

While the overall performance is low compared to GPT-based models, the trend of improvements achieved by SYNTRA remains consistent (Table 5).

We also evaluate performance on the unfiltered Playgol dataset using a wider range of LLMs, including gemma-3-27b-it, Claude Sonnet 4, and DeepSeek-V3-0324. In this setting, we use the same LLM as both the synthesis model and the transductive model.

Table 6: Task accuracies (%) of different approaches and LLMs on the unfiltered Playgol dataset.

| Approach | gemma-3-27b-it | Claude Sonnet 4 | DeepSeek-V3-0324 |
|---|---|---|---|
| AGA | 66.5 | 82.8 | 80.0 |
| AGA + SYNTRA | **72.0** | **89.8** | **90.2** |

While absolute accuracy varied across model types, we consistently observed that SYNTRA improves performance.

## C.2 Program Diversity

Table 7: Comparison of program diversity.

| Approach | Playgol | MBPP+ |
|---|---|---|
| IID (Section 5.3) | 4.32 | 3.29 |
| AGA | 6.65 | 7.20 |

Here, we demonstrate the semantic diversity of the programs generated by the IID and AGA methods. We define semantic diversity in terms of behavioral equivalence. The numbers in Table 7 represent the average number of programs, out of the 32 generated, that produce unique execution results on the training and test inputs. As shown, AGA significantly boosts diversity compared to IID sampling. This increased diversity raises the likelihood that a correct program is included in the program pool, thereby offering more opportunities for SYNTRA to improve final performance.

## C.3 Comparison with Majority Vote

We compared the performance of majority vote (MV) and SYNTRA. For MV, a majority vote was taken over outputs of generated programs, so there may not be a program exactly matching all the submitted outputs. In our experiment, MV does provide some improvement, but it's smaller than SYNTRA.

Table 8: Task accuracies (%) of majority vote (MV) and SYNTRA on unfiltered dataset.

| Approach | Playgol | MBPP+ | 1D-ARC |
|---|---|---|---|
| IID | 76.9 | 56.9 | 25.0 |
| IID + MV | 77.5 | 55.7 | 28.9 |
| IID + SYNTRA | **82.5** | **59.1** | **38.9** |

## C.4 Scaling Compute

Below are results when using MoC on the MBPP+ dataset with sample counts of 32, 64, and 128. In our experiments, MoC alone did not show a clear compute scaling effect, likely because (1) with as many as 128 concepts, the relevance of newly generated concepts diminished, and (2) as the number of programs increased, the ratio of incorrect programs also increased, raising the chance of a wrong guess when randomly selecting outputs. However, with SYNTRA, at least the second issue is mitigated, resulting in compute scaling benefits.

Table 9: Task accuracies (%) by the number of generated programs on unfiltered MBPP+ dataset.

| Approach | 32 | 64 | 128 |
|---|---|---|---|
| MoC | 78.1 | 80.9 | 77.8 |
| MoC + SYNTRA | **83.7** | **84.3** | **85.5** |

# D Case Study

## D.1 Successful Cases

**Example 1** The task is to extract the country name. The edge case here lies in the test input selected during the first iteration, where the state name appears between the city and country names. As a result, some programs extract the state name ("OR") instead of the country ("USA"). In this case, the transduction model correctly selected the ground truth "USA", effectively eliminating the hypotheses that extracted the state name.

```
Dataset: Playgol
Input-output pairs:
    Input: "ILP 2009, Leuven, Belgium, July 02-04, 2009"
    Output: "Belgium"
Iteration 1
    Test input: "ILP 2007, Corvallis, OR, USA, June 19-21, 2007"
    Output candidates: ["OR", "USA", ""]
    Transduction model prediction: "USA"
    GT output: "USA"
    Change in the number of hypotheses: 6 → 2
Iteration 2
    Test input: "ILP 2012, Dubrovnik, Croatia, September 17-19, 2012"
    Output candidates: ["Croatia", ""]
    Transduction model prediction: "Croatia"
    GT output: "Croatia"
    Change in the number of hypotheses: 2 → 1
```

**Example 2** In this task, the edge case arises in Iteration 2, where the challenge is how to handle situations with only one occurrence of the character to be removed. The transduction model chose to remove the single occurrence rather than leave it unchanged, which aligned with the ground truth output.

```
Dataset: MBPP+
Task description: Write a Python function to remove the first and last
    occurrence of a given character from the string.
```

```
Input-output pairs:
    Input: ["hello", "l"]
    Output: "heo"
Iteration 1
    Test input: ["xxx", "x"]
    Output candidates: ["x", ""]
    Transduction model prediction: "x"
    GT output: "x"
    Change in the number of hypotheses: 8 → 4
Iteration 2
    Test input: ["xrworlaaada", "x"]
    Output candidates: ["rworlaaada", "xrworlaaada", "worlaaada"]
    Transduction model prediction: "rworlaaada"
    GT output: "rworlaaada"
    Change in the number of hypotheses: 4 → 2
Iteration 3
    Test input: ["lo", "a"]
    Output candidates: ["ValueError('substring not found')", "lo"]
    Transduction model prediction: "lo"
    GT output: "lo"
    Change in the number of hypotheses: 2 → 1
```

## D.2 Failed Cases

**Example 1**   In this problem, it is difficult to use world knowledge to resolve uncertainty. The correct program logic is to output the substring up to (but not including) the first uppercase letter. However, based on the given training example alone, a program that outputs the first three characters of the input string could also satisfy it. Since the input strings in this problem are meaningless and arbitrary, there is little information available to determine which of the two programs is correct. In such cases, it would be preferable to query the user in order to generate a program that aligns with their intent.

```
Dataset: Playgol
Input-output pairs:
    Input: "worCiqshrbrgrplzaaBirqvwic"
    Output: "wor"
Iteration 1
    Test input: "htvpAsgrwbsoeigjvtryhtfp"
    Output candidates: ["htv", "", "htvp"]
    Transduction model prediction: "htv"
    GT output: "htvp"
    Change in the number of hypotheses: 3 → 1
```

**Example 2**   This is a case where the ambiguity present in the task description is reflected in the hypothesis class.

```
Dataset: MBPP+
Task description: Write a function that checks whether a string contains
    the "a" character followed by two or three "b" characters.
Input-output pairs:
    Input: "ac"
    Output: False
Iteration 1
    Test input: ""
    Output candidates: [True, False, None]
    Transduction model prediction: False
    GT output: False
    Change in the number of hypotheses: 5 -> 2
Iteration 2
    Test input: "abbbba"
```

```
Output candidates: [False, True]
Transduction model prediction: False
GT output: True
Change in the number of hypotheses: 2 -> 1
```

