# OpenReview forum: "Program Synthesis via Test-Time Transduction"
_NeurIPS.cc/2025/Conference — NeurIPS 2025 poster_

### Official Review · Reviewer_Sgwn · 2025-05-31

**Clarity:** 3
**Significance:** 3
**Originality:** 4
**Rating:** 5
**Confidence:** 5

**Summary:**

The paper considers the problem of generating programs given input-output examples (and optionally natural language). The general idea is to sample different candidate programs, and then decide between them by asking an LLM to transduce the right output for holdout tests, and preferring programs that yield outputs that align with the transductive predictions. They evaluate on text editing and MBPP problems, and consider both closed and open models.

The primary contribution of the paper is its creative method. This is an original and interesting way of leveraging holdout testcases, and of combining inductive and transductive reasoning. Because the transductive queries are expensive, they also have a clever but simple way of picking which test to try next, which is inspired by active learning criteria and works well in practice.

**Questions:**

Why do you set the temperature of the transduction model to be nonzero, given that you only call it once?

**Ethical Concerns:**

["NO or VERY MINOR ethics concerns only"]

**Final Justification:**

I'm very impressed with the new experiments on the extra domains, which shows that it can be applied to more challenging problems, and that it is not going to be viewed as niche for a NeurIPS audience. I've raised my score and think it would be a valuable contribution to the proceedings.

I hope that the authors choose a more informative title however, like "Program Synthesis with Hold-Out Transduction" or "Program Synthesis with Test-Example Transduction", or just anything which emphasizes that this is transduction on the test examples.

**Limitations:**

There could be a better discussion of weaknesses. Appendix C gives examples of failure modes, but would be better to synthesize a summary in the main text of the weaknesses. An obvious one is that you need access to holdout tests.

**Paper Formatting Concerns:**

No major formatting issues, but see nitpicks

**Quality:**

3

**Strengths And Weaknesses:**

Strengths:
1. Creativity of the method
2. Simple enough to implement in an afternoon, but reasonably technically sophisticated
3. Thorough evaluation, considering many different dimensions upon which to measure the system, and a range of reasonable ablations and variations (but see weakness 1)
4. Practicality of the method. Programming-by-Examples is a real problem, and in the real world, you do actually have holdout tests that you have access to ahead of time

Weakness: (**Note every one of these weaknesses was satisfactorily addressed during the rebuttal**)
1. The program synthesis tasks that they evaluate on are frankly not very difficult
2. They consider OpenAI models and Llama models, but could also evaluate others such as Claude, DeepSeek, etc
3. I worry that this paper could be a little bit niche for NeurIPS. Overall I think their focus on Programming-by-Examples is fine for NeurIPS, but it's borderline, particularly given weakness (1).

nitpicks:
1. eq 3: I think you mean $\hat{y}$ instead of $\tilde{y}$
2. The name Autoregressively Generated Algorithms seems gratuitous. Why not just call them programs?
3. The figure placement could be better; it's best to have figures slightly after/beside the text that discusses them.
4. The title is way too short and uninformative.

**Note to authors**: I'm currently scoring this as borderline accept but if you're willing to fix some of the issues that I raise then I'm willing to change it to accept.

---

> ### Author Rebuttal · Authors · 2025-07-31
>
> Thank you for the thoughtful review and support for our work. Below we respond to your main points.
>
> > The program synthesis tasks that they evaluate on are frankly not very difficult
>
> We agree that Playgol and MBPP+ might be relatively easy for the most capable LLMs available today. Below, we provide results on more challenging program synthesis tasks: 1D-ARC [1] and programmatic world modelling on MiniGrid [2].
>
> 1D-ARC is a difficult benchmark, with task accuracy of only 25% even when generating 128 programs. Below are the results using the MoC [3] methodology for 128 generated programs, where we can see that applying SYNTRA significantly improves performance.
>
> Results on filtered dataset
>
> | 1D-ARC (1 train / 3 test) | task acc. | example acc. | # transductive call |
> | --- | --- | --- | --- |
> | Random program | 24.0 | 28.7 | - |
> | LLM direct transduction | 41.9 | 68.1 | 372 |
> | SYNTRA w/ random query | **71.8** | **82.1** | 179 |
> | SYNTRA w/ maximin | **71.8** | 80.8 | **159** |
>
> Below are the full dataset results when using MoC, AGA, and IID as synthesis models, each with and without SYNTRA. Again, you can see SYNTRA provides a large performance boost.
>
> Results on full dataset
>
> |  | 1D-ARC |
> | --- | --- |
> | MoC | 16.7 |
> | MoC + SYNTRA | **49.4** |
> | IID | 25.0 |
> | IID + SYNTRA | 38.9 |
> | AGA | 23.9 |
> | AGA + SYNTRA | 37.8 |
>
> Next, we validate SYNTRA’s ability on programmatic world modelling (e.g. WorldCoder [4])–a complex task that requires modeling interaction mechanisms between different entities and actions. We used two MiniGrid environments (DoorKey, UnlockPickup), and focused on generating a transition function that, given the current state and action, outputs the next state.
>
> In our experiment, the synthesis model receives the current state, action list, and natural language mission as input, and generates the world models. The transduction model’s role is to select the most plausible next state candidate among multiple world model predictions. For evaluation, given a state and an action, the world model selected by SYNTRA predicts the next state, which we then compare to the ground truth next state. The state and action pairs for evaluation are collected from human play. This task is well-suited for transductive program synthesis, since the action space is typically known beforehand and can serve as a visible test input.
>
> Both the synthesis and transduction models are gpt-4.1-mini. We sampled 16 IID programs per state, and used example-level accuracy to compute transition function accuracy. Again, SYNTRA was found to provide substantial benefit for the world model synthesis task as well.
>
> |  | DoorKey | UnlockPickup |
> | --- | --- | --- |
> | IID | 57.1 | 62.9 |
> | IID + SYNTRA | **68.8** | **67.6** |
>
> [1] Xu et al. (2023), LLMs and the Abstraction and Reasoning Corpus: Successes, Failures, and the Importance of Object-based Representations, TMLR.
>
> [2] Chevalier-Boisvert et al. (2023), Minigrid & Miniworld: Modular & Customizable Reinforcement Learning Environments for Goal-Oriented Tasks, NeurIPS D&B.
>
> [3] Lee et al. (2025), Generating Diverse Hypotheses for Inductive Reasoning, NAACL.
>
> [4] Tang et al. (2024), WorldCoder, a Model-Based LLM Agent: Building World Models by Writing Code and Interacting with the Environment, NeurIPS.
>
> > They consider OpenAI models and Llama models, but could also evaluate others such as Claude, DeepSeek, etc
>
> Below are results for gemma-3-27b-it, claude 4 sonnet, and DeepSeek-V3-0324 on the unfiltered full dataset. We observe consistent performance improvements regardless of model size or type.
>
> |  | gemma-3-27b-it | claude 4 sonnet | DeepSeek-V3-0324 |
> | --- | --- | --- | --- |
> | AGA | 66.5 | 82.8 | 80.0 |
> | AGA + SYNTRA | 72.0 | 89.8 | 90.2 |
>
> > I worry that this paper could be a little bit niche for NeurIPS
>
> As the experiment results on 1D-ARC and programmatic world model synthesis tasks above demonstrate, transductive program synthesis and SYNTRA can be applied to a diverse range of tasks beyond classical PBE tasks. In 1D-ARC, the pattern recognition strengths of transduction provided substantial benefits in visual inductive reasoning tasks. In MiniGrid, SYNTRA enabled learning a more accurate world model, which would likely result in more sample-efficient planning or policy learning. We believe this has appeal beyond the program synthesis community, extending to the inductive reasoning and RL communities. Furthermore, it is worth noting that PBE, in the sense of building predictive models from data, is a general machine learning paradigm. In the real world, some tasks are more efficiently solved with programs, while others are suited to neural networks. From this perspective, the method proposed in our paper is general and applicable to a wide range of areas.
>
> > eq 3: I think you mean $\hat{y}$ instead of $\tilde{y}$
>
> Thank you. We also found the same mislabeling at L163, and will fix it accordingly.
>
> > The name Autoregressively Generated Algorithms seems gratuitous. Why not just call them programs?
>
> We felt a distinction was needed between sampling programs iid using the same prompt, and this particular methodology. For clarification: “autoregressively” here does not mean the tokens themselves are generated autoregressively, but rather that multiple algorithms (natural language form) are generated sequentially in a single LLM session, which results in more diversity among the generated programs.
>
> > The figure placement could be better.
>
> Thank you. We will incorporate this feedback.
>
> > The title is way too short and uninformative.
>
> With the title, we wished to emphasize that we are proposing a new task, rather than a method.
>
> > Why do you set the temperature of the transduction model to be nonzero, given that you only call it once?
>
> This was purely a performance-based choice. The final performance was better with a temperature of 0.7 compared to 0. Setting temperature >0 even when generating only a single response is also commonly observed to yield better results in reasoning tasks.
>
> > There could be a better discussion of weaknesses. Appendix C gives examples of failure modes, but would be better to synthesize a summary in the main text of the weaknesses.
>
> Thank you for the suggestion. We will add a dedicated limitations section. It will cover the following points:
>
> - The assumption that visible test inputs exist is necessary.
> - The method is less effective in domains/problems where the inputs are completely meaningless, making it hard for the LLM to leverage its world knowledge.
> - SYNTRA selects the optimal program from mixtures of correct and incorrect programs, but does not enhance the ability to synthesize truly complex programs for hard problems.
> - Since LLMs are used as transduction models, any undesirable biases present in the LLM may be reflected in the final output.

---

> ### Comment · Reviewer_Sgwn · 2025-08-04
>
> Having the extra domains (1D ARC and world modelling) really pushes the paper over the threshold. Thanks for your effort, and good luck.
>
> I still think the title should be changed. How about "Program Synthesis with Hold-Out Transduction"? or "Program Synthesis with Test-Example Transduction"

---

> > ### Author Response · Authors · 2025-08-06
> >
> > Thank you once again for your thoughtful review and kind words. Regarding the title, we will seriously consider making it more informative. We are currently considering “Program Synthesis with Test-Example Transduction” or “Transductive Program Synthesis with Test-Time Inputs.”

---

### Official Review · Reviewer_gBJE · 2025-06-28

**Clarity:** 3
**Significance:** 3
**Originality:** 3
**Rating:** 5
**Confidence:** 3

**Summary:**

The paper proposes a program synthesis framework called **SYNTRA** (Synthesis-by-Transduction) for tackling programming-by-example problems. SYNTRA first gathers all synthesized candidate programs that pass the given training examples. It then obtains deduplicated test output candidates by executing these programs. Finally, it eliminates candidate programs by querying an LLM to determine which input-output pair appears most consistent with the training examples. SYNTRA repeats this process for several rounds until only one program remains. The paper conducts experiments on the Playgol and MBPP+ benchmarks and achieves significant improvements compared to baseline methods.

**Questions:**

- In each round, when the transduction model makes a prediction based on the input-output pairs, does it tend to favor the output that appears most frequently among the candidates?
- What is the average number of candidate programs that pass all the training examples but produce different outputs on the test inputs for the Playgol and MBPP+ benchmarks?

**Ethical Concerns:**

["NO or VERY MINOR ethics concerns only"]

**Final Justification:**

I am raising my score from 4 to 5 for the following reasons:

1. The paper proposes a conceptually neat, scalable, and generalizable approach for reliably selecting the correct program from among plausible candidates that pass all training examples.

2. The authors have expanded their evaluation in two key ways:
- They tested the method across various state-of-the-art (SOTA) and smaller LLMs, strengthening the significance and robustness of the experimental results.
- They included two additional benchmarks: 1D-ARC (a PBE benchmark) and MiniGrid (a world-modeling benchmark), both of which showed significant performance gains, further demonstrating the method’s generalization ability and effectiveness.

3. The method achieves notable performance improvements over other widely used baseline approaches.

**Limitations:**

yes

**Paper Formatting Concerns:**

good

**Quality:**

3

**Strengths And Weaknesses:**

**Strength**

- **Quality:**

  The paper proposes a conceptually neat, scalable, and generalizable approach to reliably select the correct program from among plausible candidates that pass all the training examples. It offers an intuitive way to leverage test inputs when the training examples alone are insufficient to disambiguate among hypotheses, mirroring how humans often solve problems using both training and test examples.

- **Significance:**

  - SYNTRA significantly improves performance on the Playgol benchmark, compared to the randomly selected program, increasing accuracy from 66.6% to 93.3%, and also boosts MBPP+ performance from 70.6% to 75.2%.
  - Furthermore, as more test inputs are provided, SYNTRA’s performance continues to improve on unseen test datasets, demonstrating strong scalability and generalization.

- **Clarity:**

  The paper is well-written, easy to follow, and clearly explains the methodology and results.

**Weakness**

- The paper only conducts experiments using the GPT-4o-mini model for program synthesis and GPT-4o-mini and GPT-4.1 for transduction. Including results with different model types and sizes would strengthen the paper’s conclusions.
- Evaluating SYNTRA on a broader range of programming-by-example tasks would provide a more compelling and comprehensive understanding of its effectiveness.
- Comparing against a majority-vote baseline would also be a valuable addition.
- It would be interesting to analyze how SYNTRA scales with increased compute during the program synthesis stage, particularly when generating more diverse hypotheses and a larger pool of candidate programs.

---

> ### Author Rebuttal · Authors · 2025-07-31
>
> Thank you for the thoughtful review and suggestions for improvements. Below we respond to your main points.
>
> > The paper only conducts experiments using the GPT-4o-mini model for program synthesis and GPT-4o-mini and GPT-4.1 for transduction. Including results with different model types and sizes would strengthen the paper’s conclusions.
>
> Appendix B.1 includes experiments using the smaller open-source models Llama-3.1-8B-Instruct and Llama-3.1-70B-Instruct. Additionally, results for gemma-3-27b-it, claude 4 sonnet, and DeepSeek-V3-0324 are provided below. These results are for the unfiltered full dataset. Consistent performance improvements are observed regardless of model size or type.
>
> |  | gemma-3-27b-it | claude 4 sonnet | DeepSeek-V3-0324 |
> | --- | --- | --- | --- |
> | AGA | 66.5 | 82.8 | 80.0 |
> | AGA + SYNTRA | 72.0 | 89.8 | 90.2 |
>
> > Evaluating SYNTRA on a broader range of programming-by-example tasks would provide a more compelling and comprehensive understanding of its effectiveness.
>
> To validate our methodology on a broader range of program synthesis tasks, we present performance results on two challenging tasks, 1D-ARC [1] and programmatic world modelling on MiniGrid [2].
>
> 1D-ARC is a PBE benchmark that requires reasoning skills more related to human cognitive priors. It is a more difficult benchmark, with a task accuracy of only 25% even when generating 128 programs per question. Below are the results using the MoC [3] method, generating 128 programs per task, with and without the application of SYNTRA. You can see that applying SYNTRA significantly improves performance.
>
> Results on filtered dataset
>
> | 1D-ARC (1 train / 3 test) | task acc. | example acc. | # transductive call |
> | --- | --- | --- | --- |
> | Random program | 24.0 | 28.7 | - |
> | LLM direct transduction | 41.9 | 68.1 | 372 |
> | SYNTRA w/ random query | **71.8** | **82.1** | 179 |
> | SYNTRA w/ maximin | **71.8** | 80.8 | **159** |
>
> Below are the full dataset results when using MoC, AGA, and IID as synthesis models, each with and without SYNTRA. Again, you can see SYNTRA provides a large performance boost.
>
> Results on full dataset
>
> |  | 1D-ARC |
> | --- | --- |
> | MoC | 16.7 |
> | MoC + SYNTRA | **49.4** |
> | IID | 25.0 |
> | IID + SYNTRA | 38.9 |
> | AGA | 23.9 |
> | AGA + SYNTRA | 37.8 |
>
> Next, we validate SYNTRA’s ability on programmatic world modelling (e.g. WorldCoder [4])–a complex task that requires modeling interaction mechanisms between different entities and actions. We used two MiniGrid environments (DoorKey, UnlockPickup), and focused on generating a transition function that, given the current state and action, outputs the next state.
>
> In our experiment, the synthesis model receives the current state, action list, and natural language mission as input, and generates the world models. The transduction model’s role is to select the most plausible next state candidate among multiple world model predictions. For evaluation, given a state and an action, the world model selected by SYNTRA predicts the next state, which we then compare to the ground truth next state. The state and action pairs for evaluation are collected from human play. This task is well-suited for transductive program synthesis, since the action space is typically known beforehand and can serve as a visible test input.
>
> Both the synthesis and transduction models are gpt-4.1-mini. We sampled 16 IID programs per state, and used example-level accuracy to compute transition function accuracy. Again, SYNTRA was found to provide substantial benefit for the world model synthesis task as well.
>
> |  | DoorKey | UnlockPickup |
> | --- | --- | --- |
> | IID | 57.1 | 62.9 |
> | IID + SYNTRA | **68.8** | **67.6** |
>
> [1] Xu et al. (2023), LLMs and the Abstraction and Reasoning Corpus: Successes, Failures, and the Importance of Object-based Representations, TMLR.
>
> [2] Chevalier-Boisvert et al. (2023), Minigrid & Miniworld: Modular & Customizable Reinforcement Learning Environments for Goal-Oriented Tasks, NeurIPS D&B.
>
> [3] Lee et al. (2025), Generating Diverse Hypotheses for Inductive Reasoning, NAACL.
>
> [4] Tang et al. (2024), WorldCoder, a Model-Based LLM Agent: Building World Models by Writing Code and Interacting with the Environment, NeurIPS.
>
> > Comparing against a majority-vote baseline would also be a valuable addition.
>
> We compared the performance of majority vote (self-consistency, SC) and SYNTRA. For SC, a majority vote was taken over outputs of generated programs, so there may not be a program exactly matching all the submitted outputs. In our experiment, SC does provide some improvement, but it’s smaller than SYNTRA.
>
> |  | Playgol | MBPP+ | 1D-ARC |
> | --- | --- | --- | --- |
> | IID | 76.9 | 56.9 | 25.0 |
> | IID + SC | 77.5 | 55.7 | 28.9 |
> | IID + SYNTRA | **82.5** | **59.1** | **38.9** |
>
> > It would be interesting to analyze how SYNTRA scales with increased compute during the program synthesis stage, particularly when generating more diverse hypotheses and a larger pool of candidate programs.
>
> Below are results when using MoC on the MBPP+ dataset with sample counts of 32, 64, and 128. In our experiments, MoC alone did not show a clear compute scaling effect, likely because (1) with as many as 128 concepts, the relevance of newly generated concepts diminished, and (2) as the number of programs increased, the ratio of incorrect programs also increased, raising the chance of a wrong guess when randomly selecting outputs. However, with SYNTRA, at least the second issue is mitigated, resulting in compute scaling benefits.
>
> |  | 32 | 64 | 128 |
> | --- | --- | --- | --- |
> | MoC | 78.1 | 80.9 | 77.8 |
> | MoC + SYNTRA | **83.7** | **84.3** | **85.5** |
>
> > In each round, when the transduction model makes a prediction based on the input-output pairs, does it tend to favor the output that appears most frequently among the candidates?
>
> Since the output candidates are deduplicated before being given to the model, each output only appears once, so this does not happen.
>
> > What is the average number of candidate programs that pass all the training examples but produce different outputs on the test inputs for the Playgol and MBPP+ benchmarks?
>
> The requested information corresponds to the cardinality of the initial version space $\mathcal{V}_0$. Below are the average cardinalities of $\mathcal{V}_0$ for Playgol, MBPP+, and the newly added 1D-ARC dataset. All use AGA, and the numbers in parentheses indicate the number of generated programs.
>
> |  | Playgol | MBPP+ | 1D-ARC |
> | --- | --- | --- | --- |
> | Avg. cardinality of $\mathcal{V}_0$ | 3.21 (32) | 2.96 (32) | 9.65 (128) |

---

> > ### Comment · Reviewer_gBJE · 2025-08-04
> >
> > I thank the authors for their response. The clarification and additional results provided have addressed my concerns, and I will raise my score to 5 (accept) in support.

---

### Official Review · Reviewer_vRSo · 2025-07-02

**Clarity:** 3
**Significance:** 2
**Originality:** 2
**Rating:** 4
**Confidence:** 3

**Summary:**

This paper introduces transductive program synthesis, a new task.  And this paper proposes a framework, SYNTRA, that treats program synthesis as an active learning problem over a finite set of hypotheses, where each hypothesis corresponds to the outputs of a candidate program on the test set. To efficiently find the correct solution, SYNTRA uses a transductive LLM to predict the output for each test input, which is selected using a greedy maximin algorithm to minimize the number of LLM queries. This method demonstrated significant improvements in accuracy compared to traditional inductive and transductive techniques on MBPP and Playgol.

**Questions:**

Please see my comments above.

**Ethical Concerns:**

["NO or VERY MINOR ethics concerns only"]

**Final Justification:**

I am convinced by the authors' rebuttal. Indeed, selecting the correct program among a mix of correct and incorrect programs is itself an important contribution. And the author's privide additional experiments to address my concern.

**Limitations:**

yes

**Quality:**

2

**Strengths And Weaknesses:**

Strength:
1. The code is open-sourced, which helps replication.
2. The paper is well-written and easy to follow.
3. This paper introduces a new task, transductive program synthesis, which I think is indeed valuable and interesting for real-world scenarios.
4. According to the experimental results, the proposed framework is effective on accuracy.

Weakness:
1. There is a strong assumption behind the proposed framework. That is, there is at least one correct program in the initially generated sets. If there is not, all of the computations in this framework are meaningless.
2. For transductive models, it highly relies on the generated pseudo-label to remove hypotheses. Similar to weakness 1, this can not be mitigated if the pseudo-label is wrong.
3. The reason why the above two limitations do not have a significant negative influence on the framework performance is that the two evaluated benchmarks are too easy for current LLMs. (At least I am quite familiar with MBPP, and I am sure it is too easy for GPT-4o or GPT-4o mini that are used for evaluation). I suggest that the authors should evaluate the proposed framework on more challenging benchmarks.
4. Although the authors reported in Table 1 that there are fewer # $\tau$ Calls, the authors do not discuss the # $\sigma$ Calls compared with the baseline methods.

---

> ### Author Rebuttal · Authors · 2025-07-31
>
> Thank you for the thoughtful review and suggestions for improvements. Below we respond to your main points.
>
> > There is a strong assumption behind the proposed framework. That is, there is at least one correct program in the initially generated sets. If there is not, all of the computations in this framework are meaningless.
>
> We agree that our methodology assumes the existence of at least one correct program. However, selecting the correct program among a mix of correct and incorrect programs is itself an important contribution, because such situations are quite common in practice. As can be seen in the performances on Table 2 and other benchmarks presented below, SYNTRA substantially boosts performance calculated across full dataset (including those where there are no correct programs at all). Moreover, even if there is no correct program, SYNTRA can still find good programs that are correct for many inputs, thereby improving example-level accuracy. We would also like to emphasize that our paper is the first to propose the task of transductive program synthesis and a system to address it, which we hope will inspire a variety of future research directions aimed at improving this approach. We believe that exploring methods to mitigate the issue you raised would be a promising direction for future research.
>
> > For transductive models, it highly relies on the generated pseudo-label to remove hypotheses. Similar to weakness 1, this can not be mitigated if the pseudo-label is wrong.
>
> We agree that our methodology depends on the accuracy of pseudo-labels. However, the performances in Table 1, Table 2, and on more challenging benchmarks below empirically show that these pseudo-labels are quite accurate.
>
> As a potential solution to this issue, one could perform multiple samplings and use a majority vote as the pseudo-label. Alternatively, instead of using a hard label, it might be possible to use a soft label, which could provide opportunities to recover from errors. For example, in the transduction model, after sampling pseudo-labels multiple times, one could train a light-weight model with these pseudo-labels to obtain a distribution over the hypotheses.
>
> > The reason why the above two limitations do not have a significant negative influence on the framework performance is that the two evaluated benchmarks are too easy for current LLMs. I suggest that the authors should evaluate the proposed framework on more challenging benchmarks.
>
> As empirical evidence that the above two limitations do not have a significant negative influence even on harder benchmarks, we present results on challenging program synthesis tasks such as 1D-ARC [1] and programmatic world modelling on MiniGrid [2].
>
> 1D-ARC is a difficult benchmark, with a task accuracy of only 25% even when generating 128 programs per question. Below are the results using the MoC [3] method, generating 128 programs per task, with and without the application of SYNTRA. You can see that applying SYNTRA significantly improves performance.
>
> Results on filtered dataset
>
> | 1D-ARC (1 train / 3 test) | task acc. | example acc. | # transductive call |
> | --- | --- | --- | --- |
> | Random program | 24.0 | 28.7 | - |
> | LLM direct transduction | 41.9 | 68.1 | 372 |
> | SYNTRA w/ random query | **71.8** | **82.1** | 179 |
> | SYNTRA w/ maximin | **71.8** | 80.8 | **159** |
>
> Below are the full dataset results when using MoC, AGA, and IID as synthesis models, each with and without SYNTRA. Again, you can see SYNTRA provides a large performance boost.
>
> Results on full dataset
>
> |  | 1D-ARC |
> | --- | --- |
> | MoC | 16.7 |
> | MoC + SYNTRA | **49.4** |
> | IID | 25.0 |
> | IID + SYNTRA | 38.9 |
> | AGA | 23.9 |
> | AGA + SYNTRA | 37.8 |
>
> Next, we validate SYNTRA’s ability on programmatic world modelling (e.g. WorldCoder [4])–a complex task that requires modeling interaction mechanisms between different entities and actions. We used two MiniGrid environments (DoorKey, UnlockPickup), and focused on generating a transition function that, given the current state and action, outputs the next state.
>
> In our experiment, the synthesis model receives the current state, action list, and natural language mission as input, and generates the world models. The transduction model’s role is to select the most plausible next state candidate among multiple world model predictions. For evaluation, given a state and an action, the world model selected by SYNTRA predicts the next state, which we then compare to the ground truth next state. The state and action pairs for evaluation are collected from human play. This task is well-suited for transductive program synthesis, since the action space is typically known beforehand and can serve as a visible test input.
>
> Both the synthesis and transduction models are gpt-4.1-mini. We sampled 16 IID programs per state, and used example-level accuracy to compute transition function accuracy. Again, SYNTRA was found to provide substantial benefit for the world model synthesis task as well.
>
> |  | DoorKey | UnlockPickup |
> | --- | --- | --- |
> | IID | 57.1 | 62.9 |
> | IID + SYNTRA | **68.8** | **67.6** |
>
> [1] Xu et al. (2023), LLMs and the Abstraction and Reasoning Corpus: Successes, Failures, and the Importance of Object-based Representations, TMLR.
>
> [2] Chevalier-Boisvert et al. (2023), Minigrid & Miniworld: Modular & Customizable Reinforcement Learning Environments for Goal-Oriented Tasks, NeurIPS D&B.
>
> [3] Lee et al. (2025), Generating Diverse Hypotheses for Inductive Reasoning, NAACL.
>
> [4] Tang et al. (2024), WorldCoder, a Model-Based LLM Agent: Building World Models by Writing Code and Interacting with the Environment, NeurIPS.
>
> > Although the authors reported in Table 1 that there are fewer # $\tau$ Calls, the authors do not discuss the # $\sigma$ Calls compared with the baseline methods.
>
> The reason # $\sigma$ Calls were not reported in the table is because the number of synthesis model calls is the same for all approaches—40 per problem (8 for AGA’s algorithm generation, 32 for code generation). We will make this point clearer in the experiment section. In addition, a more detailed discussion of the costs of $\sigma$ and $\tau$ calls and compute cost can be found in L355–L363.

---

> > ### Author Response · Authors · 2025-08-06
> >
> > We sincerely appreciate your participation in reviewing our research and the valuable feedback you provided. We would be grateful if you could review our rebuttal and let us know at your convenience whether we have adequately addressed your concerns.

---

> > > ### Comment · Reviewer_vRSo · 2025-08-09
> > >
> > > I thank the authors for their clarifications. And I am sorry for the late reply. I was convinced by the rebuttal. Thus, I will raise my score to 4. Please add the rebuttal content to the camera-ready version.

---

### Official Review · Reviewer_2PpF · 2025-07-03

**Clarity:** 3
**Significance:** 2
**Originality:** 3
**Rating:** 5
**Confidence:** 3

**Summary:**

This method proposes a new code generation setting, in which in addition to a pronlem description, a few test inputs are known (but not the golden output). They propose to leverage outputs from running the code snippet on these test inputs, by having the LLM select which outputs it considers the most well-fitting to the original description,  as a semi-symbolic method to increase performance of LLM generated code.

**Questions:**

- Did the authors experiment with the LLM just generating additional test inputs by itself? This seems like a trivial way to potentially increase performance even on non-transduction datasets and might further strengthen the method (also cf Code Agents which build their own test cases to cross-check their code repairs).
- The main benefit of your maximin approach to hypothesis elimination appears to be reduction of LLM queries. I would assume that the corresponding queries are quite cheap and random queries sometimes outperform maximin - so did you consider any other order of querying that might maximize the likelihood of picking the best matching hypothesis (trading off queries)?
- Table 1: how does your method compare to other approaches to trade of queries vs performance such as CoT/reasoning, self-consistency, etc? I think this would be interesting to show as well.
- Did you also try other models than GPT-4o and 4.1? I think it would be especially interesting to see if this can improve performance of weaker models or introduces additional confusion.

**Ethical Concerns:**

["NO or VERY MINOR ethics concerns only"]

**Final Justification:**

The main concern of myself and the other reviewers was that this papers evaluation of models and datasets was insufficient as thorough evidence for the effectiveness of the method. This was adequately addressed by additional experiments during the rebuttal. All other minor issues have been resolved as well.

**Limitations:**

Yes

**Quality:**

3

**Strengths And Weaknesses:**

I think the idea presented in the paper is interesting and original. The paper is well-structured and overall clearly written.

Regarding the evaluation, I have a number of points.
- MBPP+ appears to be almost solved by the method presented. You even report that you had to filter out samples where the models consistently generate correct solutions (Line 227), resulting in a rather small dataset. Maybe you can instead try more complicated benchmarks such as Rust programs [1,2] or LBPP [3]
- The formulation in Line 233/234 is a bit unclear: does "predicted correctly" refer to the LLM predicting the output correctly or the resulting program computing the correct output?
- In Figure 4, it seems that increasing test case as inputs may result in decreasing performance. This appears unintuitive and not discussed - do you have an intuition on why this is happening? Is the model more likely to wrongly mark test outputs?
- A general question is how diverse the test inputs to the MBPP test cases are. Do 40 test inputs generally already cover all relevant edge cases? Do the remaining 10 unseen test cases cover relevant edge cases? Can you somehow quantify this by i.e., measuring the line coverage of each test input on the golden solution and checking whether all test inputs cover all lines?

[1] BabelCode MBPP (https://huggingface.co/datasets/gabeorlanski/bc-mbpp)
[2] Khatry et al., "CRUST-Bench: A Comprehensive Benchmark for C-to-safe-Rust Transpilation", arXiv 2025
[3] Matton et al., "On Leakage of Code Generation Evaluation Datasets", EMNLP 2024

---

> ### Author Rebuttal · Authors · 2025-07-31
>
> Thank you for the thoughtful review and suggestions for improvements. Below we respond to your main points.
>
> > MBPP+ appears to be almost solved by the method presented. You even report that you had to filter out samples where the models consistently generate correct solutions (Line 227), resulting in a rather small dataset. Maybe you can instead try more complicated benchmarks.
>
> Thank you for the suggestion. Below, we present performance results on two challenging tasks, 1D-ARC [1] and programmatic world modelling on MiniGrid [2].
>
> 1D-ARC is a difficult benchmark, with a task accuracy of only 25% even when generating 128 programs per question. Below are the results using the MoC [3] method, generating 128 programs per task, with and without the application of SYNTRA. You can see that applying SYNTRA significantly improves performance.
>
> Results on filtered dataset
>
> | 1D-ARC (1 train / 3 test) | task acc. | example acc. | # transductive call |
> | --- | --- | --- | --- |
> | Random program | 24.0 | 28.7 | - |
> | LLM direct transduction | 41.9 | 68.1 | 372 |
> | SYNTRA w/ random query | **71.8** | **82.1** | 179 |
> | SYNTRA w/ maximin | **71.8** | 80.8 | **159** |
>
> Below are the full dataset results when using MoC, AGA, and IID as synthesis models, each with and without SYNTRA. Again, you can see SYNTRA provides a large performance boost.
>
> Results on full dataset
>
> |  | 1D-ARC |
> | --- | --- |
> | MoC | 16.7 |
> | MoC + SYNTRA | **49.4** |
> | IID | 25.0 |
> | IID + SYNTRA | 38.9 |
> | AGA | 23.9 |
> | AGA + SYNTRA | 37.8 |
>
> Next, we validate SYNTRA’s ability on programmatic world modelling (e.g. WorldCoder [4])–a complex task that requires modeling interaction mechanisms between different entities and actions. We used two MiniGrid environments (DoorKey, UnlockPickup), and focused on generating a transition function that, given the current state and action, outputs the next state.
>
> In our experiment, the synthesis model receives the current state, action list, and natural language mission as input, and generates the world models. The transduction model’s role is to select the most plausible next state candidate among multiple world model predictions. For evaluation, given a state and an action, the world model selected by SYNTRA predicts the next state, which we then compare to the ground truth next state. The state and action pairs for evaluation are collected from human play. This task is well-suited for transductive program synthesis, since the action space is typically known beforehand and can serve as a visible test input.
>
> Both the synthesis and transduction models are gpt-4.1-mini. We sampled 16 IID programs per state, and used example-level accuracy to compute transition function accuracy. Again, SYNTRA was found to provide substantial benefit for the world model synthesis task as well.
>
> |  | DoorKey | UnlockPickup |
> | --- | --- | --- |
> | IID | 57.1 | 62.9 |
> | IID + SYNTRA | **68.8** | **67.6** |
>
> [1] Xu et al. (2023), LLMs and the Abstraction and Reasoning Corpus: Successes, Failures, and the Importance of Object-based Representations, TMLR.
>
> [2] Chevalier-Boisvert et al. (2023), Minigrid & Miniworld: Modular & Customizable Reinforcement Learning Environments for Goal-Oriented Tasks, NeurIPS D&B.
>
> [3] Lee et al. (2025), Generating Diverse Hypotheses for Inductive Reasoning, NAACL.
>
> [4] Tang et al. (2024), WorldCoder, a Model-Based LLM Agent: Building World Models by Writing Code and Interacting with the Environment, NeurIPS
>
> > does "predicted correctly" refer to the LLM predicting the output correctly or the resulting program computing the correct output?
>
> It refers to the latter. We will revise this part to make it clearer.
>
> > In Figure 4, it seems that increasing test case as inputs may result in decreasing performance.
>
> We believe the variability shown in Figure 4 is simply due to differences between random seeds. There is not a consistent trend of decrease, as in some cases performance remains similar or even rises again depending on the approach, so it is not consistent. However, if we were to name a possible cause for performance drop, as the number of test cases increases, the number of candidate hypotheses usually increases, which may make it harder to select the correct one.
>
> > A general question is how diverse the test inputs to the MBPP test cases are.
>
> In MBPP+ [5], which we used, a large number of new test cases are created, and the test cases are reduced while maximally preserving the GT program’s branch coverage, so we believe it is likely that a sufficient number of edge cases are included.
>
> [5] Liu et al., Is Your Code Generated by ChatGPT Really Correct? Rigorous Evaluation of Large Language Models for Code Generation, NeurIPS 2023.
>
> > Did the authors experiment with the LLM just generating additional test inputs by itself?
>
> We have not tried this, but it is a very good idea! We expect performance to be somewhere between SYNTRA and the baseline, but since a visible test set would not be needed, it could be broadly applicable and is a promising method for inductive program synthesis. We will add this to the discussion.
>
> > Did you consider any other order of querying that might maximize the likelihood of picking the best matching hypothesis (trading off queries)?
>
> The most naive approach to maximizing the likelihood of picking the best matching hypothesis is to use all inputs as queries and aggregate the results via majority voting, etc. This is the most accurate, but the compute cost increases linearly with the number of tests, so it’s not optimal. Although we don’t have an immediate idea for a method that jointly optimizes cost and performance, this would be a good direction for future work.
>
> > How does your method compare to other approaches to trade off queries vs performance such as CoT/reasoning, self-consistency, etc?
>
> CoT is already used as a prompting strategy in all baselines, so we compared the performance of self-consistency (SC) and SYNTRA. For self-consistency, a majority vote was taken over outputs of generated programs, so there may not be a program exactly matching all the submitted outputs. In our experiment, SC does provide some improvement, but it’s smaller than SYNTRA.
>
> |  | Playgol | MBPP+ | 1D-ARC |
> | --- | --- | --- | --- |
> | IID | 76.9 | 56.9 | 25.0 |
> | IID + SC | 77.5 | 55.7 | 28.9 |
> | IID + SYNTRA | **82.5** | **59.1** | **38.9** |
>
> > Did you also try other models than GPT-4o and 4.1? I think it would be especially interesting to see if this can improve performance of weaker models or introduces additional confusion.
>
> In Appendix B.1, there are experiments using the smaller open source models Llama-3.1-8B-Instruct and Llama-3.1-70B-Instruct. Additionally, below are results for gemma-3-27b-it, claude 4 sonnet, and DeepSeek-V3-0324 on the full, unfiltered dataset. SYNTRA shows consistent performance gains regardless of model size and type.
>
> |  | gemma-3-27b-it | claude 4 sonnet | DeepSeek-V3-0324 |
> | --- | --- | --- | --- |
> | AGA | 66.5 | 82.8 | 80.0 |
> | AGA + SYNTRA | 72.0 | 89.8 | 90.2 |

---

> > ### Comment · Reviewer_2PpF · 2025-08-04
> >
> > I thank the authors for their additional, interesting results and clarifications.
> >
> > Regarding Figure 4, the confusion could most likely be entirely resolved by adding confidence intervals, which, should either overlap in the "random performance drop" region or confirm a hypothesis where too many examples make picking a correct solution difficult.
> >
> > It seems that the other reviewers otherwise agree on the points of dataset and model variety, which appears to me to be sufficiently  addressed by the rebuttal. I recommend the authors to include the results on the additional models and datasets prominently in the next revision of the paper and wish them good luck.

---

> > > ### Author Response · Authors · 2025-08-07
> > >
> > > Thank you once again for your thoughtful review and kind words.
> > >
> > > To ensure more rigorous verification on Table 4 results, we investigated statistics over 5 runs using different test sets. We found that task accuracy indeed shows a slight decrease as the number of test examples increases. This drop was small enough that it wasn’t clearly noticeable in the initial experiments with a single run. Additionally, the average number of hypotheses ($|\mathcal{V}_0|$) tends to increase with more test examples, while the example-level accuracy appears to remain nearly constant.
> > >
> > > The table below shows the mean and standard deviation when the number of test inputs is 5, 10, 20, and 40.
> > >
> > > |  | 5 | 10 | 20 | 40 |
> > > | --- | --- | --- | --- | --- |
> > > | Task Acc. | 88.6 ± 1.3 | 87.2 ± 1.6 | 86.6 ± 1.3 | 86.3 ± 0.9 |
> > > | Example Acc. | 95.6 ± 0.5 | 95.2 ± 0.5 | 95.4 ± 0.6 | 95.6 ± 0.4 |
> > > | Num. Hypotheses | 2.61 ± 0.06 | 2.88 ± 0.04 | 3.10 ± 0.04 | 3.26 ± 0.01 |
> > >
> > > We would like to offer one plausible hypothesis that could explain this trend. If a single extremely difficult edge case is included in the test set, additional hypotheses that differ only on that one case may be generated. If the transduction model happens to mispredict this one edge case, final hypothesis will still be correct on most test examples (thus no significant drop in example accuracy). However, since task accuracy only gives a score of 1 if all outputs are correct, the task accuracy will drop. As the number of test examples increases, so does the probability of such difficult edge cases being included, which explains the observed trend.
> > >
> > > Task accuracy is a strict metric that only rewards completely correct hypothesis across all test items. Therefore, we believe that a decrease in task accuracy as the number of test examples increases is a natural tendency that no approach can entirely avoid.

---

### Note · Authors · 2025-08-13

We thank the reviewers and ACs for their time and constructive feedback.

Our paper was generally well received: borderline accept with confidence 5 (Reviewer **Sgwn**), borderline accept with confidence 3 (Reviewer **2PpF**), borderline accept with confidence 3 (Reviewer **gBJE**), and borderline reject with confidence 3 (Reviewer **vRSo**).

Reviewers highlighted (i) the originality and interestingness of our approach, (ii) the practicality of the proposed task, and (iii) strong empirical gains.

They initially raised concerns about limited model diversity and dataset difficulty, as well as minor points (e.g., comparison with majority-vote baseline, number of test inputs).

In our rebuttal, we added experiments on harder domains (1D-ARC, MiniGrid world modeling), included more models (Gemma, Claude, DeepSeek), and provided comparisons showing SYNTRA outperforms majority vote. We also added statistics for the Figure 4 results, a compute scaling experiment, and refined presentation details. All reviewers indicated that the rebuttal satisfactorily addressed their concerns. We will incorporate all requested additions and improvements into the camera-ready version.

Post-rebuttal outcomes are as follows:
- Reviewer **Sgwn**: acknowledged our rebuttal and commented, “Having the extra domains (1D ARC and world modelling) really pushes the paper over the threshold” (initial score: 4, borderline accept)
- Reviewer **2PpF** responded to us that his questions are sufficiently addressed by the rebuttal (initial score: 4, borderline accept)
- Reviewer **gBJE** initially recommended borderline accept of our paper and increased their score to 5 (accept) after the rebuttal
- Reviewer **vRSo** initially recommended borderline reject of our paper and increased their score to 4 (borderline accept) after the rebuttal

---

### Decision · Program_Chairs · 2025-09-17

**Decision:**

Accept (poster)

**Comment:**

The paper proposes SYNTRA, a novel framework that improves robustness in program synthesis by treating synthesis as an active learning problem over a finite hypothesis class defined by candidate programs’ outputs. Test inputs are chosen via a greedy maximin algorithm to reduce the number of required LLM queries.

Strengths:
- Interesting, creative, and original idea that is scalable and generalizable for reliably selecting correct programs among candidates (2PpF, gBJE, Sgwn)
- Introduces transductive program synthesis, which is valuable and relevant for real-world applications (vRSo, Sgwn)
- Strong experimental results demonstrating effectiveness of the framework in improving accuracy (vRSo, gBJE, Sgwn)

Weaknesses:
- Benchmarks used are relatively easy; more complex benchmarks or broader programming-by-example tasks would strengthen the claims (2PpF, vRSo, Sgwn, gBJE). However, the authors added additional benchmarks during rebuttal.
- Experiments initially focused on OpenAI models only (gBJE, Sgwn); the authors already addressed this issue in rebuttal with results on gemma-3-27b-it, Claude 4 Sonnet, and DeepSeek-V3-0324.
- Relies on the strong assumption that at least one correct program exists in the initial candidate set (vRSo).

Overall, SYNTRA presents a creative and original framework with solid empirical performance and a novel perspective on program synthesis. While some assumptions and benchmark choices limit its strength, the contribution is promising and well-motivated.